# Temporal Eating Patterns and Eating Windows among Adults with Overweight or Obesity

**DOI:** 10.3390/nu13124485

**Published:** 2021-12-15

**Authors:** Collin J. Popp, Margaret Curran, Chan Wang, Malini Prasad, Keenan Fine, Allen Gee, Nandini Nair, Katherine Perdomo, Shirley Chen, Lu Hu, David E. St-Jules, Emily N. C. Manoogian, Satchidananda Panda, Mary Ann Sevick, Blandine Laferrère

**Affiliations:** 1Department of Population Health, Center for Healthful Behavior Change, New York University Langone Health, 180 Madison Ave, New York, NY 10016, USA; Margaret.Curran@nyulangone.org (M.C.); Katherine.Perdomo@nyulangone.org (K.P.); Shirley.Chen2@nyulangone.org (S.C.); lu.hu@nyulangone.org (L.H.); Mary.Sevick@nyulangone.org (M.A.S.); 2Department of Population Health, Division of Biostatistics, New York University Langone Health, 180 Madison Ave, New York, NY 10016, USA; Chan.Wang@nyulangone.org; 3Department of Medicine, Division of Endocrinology, New York Obesity Research Center, Columbia University Irving Medical Center, 1150 Saint Nicholas Avenue, R-121-G, New York, NY 10032, USA; mprasad2@gradcenter.cuny.edu (M.P.); keenan.fine@northwestern.edu (K.F.); ag4165@cumc.columbia.edu (A.G.); Nandini.Nair@nyulangone.org (N.N.); bbl14@columbia.edu (B.L.); 4Department of Nutrition, University of Nevada, Reno, 1664 N. Virginia Street, Reno, NV 89557, USA; stjules@unr.edu; 5Regulatory Biology Department, Salk Institute for Biological Studies, 10010 N Torrey Pines Rd., La Jolla, CA 92037, USA; emanoogian@salk.edu (E.N.C.M.); panda@salk.edu (S.P.); 6Department of Medicine, New York University Langone Health, 550 First Avenue, New York, NY 10016, USA

**Keywords:** meal timing, breakfast skipping, time-restricted eating, intermittent fasting, meal patterns, alternate day fasting

## Abstract

We aim to describe temporal eating patterns in a population of adults with overweight or obesity. In this cross-sectional analysis, data were combined from two separate pilot studies during which participants entered the timing of all eating occasions (>0 kcals) for 10–14 days. Data were aggregated to determine total eating occasions, local time of the first and last eating occasions, eating window, eating midpoint, and within-person variability of eating patterns. Eating patterns were compared between sexes, as well as between weekday and weekends. Participants (*n* = 85) had a median age of 56 ± 19 years, were mostly female (>70%), white (56.5%), and had a BMI of 31.8 ± 8.0 kg/m^2^. The median eating window was 14 h 04 min [12 h 57 min–15 h 21 min], which was significantly shorter on the weekend compared to weekdays (*p* < 0.0001). Only 13.1% of participants had an eating window <12 h/d. Additionally, there was greater irregularity with the first eating occasion during the week when compared to the weekend (*p* = 0.0002). In conclusion, adults with overweight or obesity have prolonged eating windows (>14 h/d). Future trials should examine the contribution of a prolonged eating window on adiposity independent of energy intake.

## 1. Introduction

The age-adjusted prevalence of obesity in adults is over 40% of the US population [1]. Obesity and poor metabolic health have been linked to chrononutrition or temporal eating patterns, which refers to the timing, frequency and regularity of food intake throughout the day [2]. Chrononutrition considers the interaction between meal timing, circadian biology, and nutrient metabolism. Disruption of the circadian rhythm caused by erratic eating patterns [3], uneven caloric distribution [4], and prolonged eating windows to include hours usually reserved for sleep, are recognized risk factors for obesity and other chronic diseases (e.g., type 2 diabetes, cardiovascular disease) [5,6].

The National Health and Nutrition Examination Survey (2009–2014) reveals that US adults report approximately five eating occasions per day, with breakfast occurring at approximately 08:00, dinner around 18:25 and the last eating occasion at 20:18 [4]. The distribution of calories is skewed toward the evening, with 45% of daily energy intake consumed with dinner and the after-dinner snack and <20% of total energy intake consumed at the breakfast meal [4]. Although intake is highly concentrated later in the day, eating-pattern analyses show the typical duration of the daily eating window is 12–15 h/d [3,4,7,8], with <10% of adults having an eating window <12 h/d [3]. While an eating-window cutoff has yet to be established, restricting the daily eating window to <12 h/d may confer metabolic health benefits, including improvements in glycemia, reductions in blood pressure, blood lipids, and body weight [9,10].

The body’s endogenous circadian system orchestrates rhythms in metabolism over a 24-h period [11,12,13]. Meal timing may act as an external cue (i.e., zeitgeber), entraining the external environment with the body’s endogenous circadian system [14]. Greater energy intake in the morning (2-h after waking) is associated with lower odds of obesity [6]; in contrast, greater consumption of energy intake later in the day is associated with obesity [6,15,16]. Intermittent fasting interventions that alter feed/fasting periods, such as time-restricted eating (TRE), may mitigate obesity and chronic disease risks [3,17,18,19,20]. However, prior studies have often neglected to assess baseline habitual eating patterns and only screened for breakfast consumption [17,21] or prior fasting [22,23]. Quantifying baseline temporal eating patterns may be important in the selection of individuals who may benefit most from chrononutrition-based interventions, such as TRE. Our aim is to describe temporal eating patterns in adults with overweight or obesity and to assess the within- and between-person variability in duration of daily eating windows in this population.

## 2. Materials and Methods

We used cross-sectional baseline data from two separate intervention pilot studies at Columbia University and New York University Langone Health. All data were collected prior to the beginning of each intervention. Ethical approvals for both studies were obtained from the respective institutional review boards of Columbia University and New York University Langone Health. All participants provided written informed consent.

### 2.1. NY-TREAT Study

The New York Time-Restricted EATing (NY-TREAT) pilot study (clincialtrials.gov NCT03956290) was designed to assess the effect of TRE in individuals with long eating windows (≥14 h/d). The first phase of this study consisted of an observational baseline cross-sectional 2-week continuous assessment of eating patterns via a smartphone app in order to identify individuals eating for ≥14 h/day. Inclusion criteria included adults ages 30–75 years, BMI 25–50 kg/m^2^, in possession of a smartphone, English-speaking, and living in the New York City geographical area. Exclusion criteria included current shift work or shift work within the last 6 months, planned travel across more than 1 time zone during the study period, significant organ-system dysfunction/disease (e.g., diabetes, severe pulmonary diagnosis, kidney or cardiovascular disease), history of seizure disorder, recent bariatric surgery (<2 years), weight-loss medication and history of significant psychiatric disorder (e.g., prior or current diagnosis).

Potential adult participants were recruited from the New York City metropolitan area via advertisements in local communities, citywide wellness programs, New York Presbyterian Hospital, the Columbia University Medical Center RecruitMe website, and NIH’s Researchmatch. Potentially eligible participants were screened via phone to assess the self-reported timing of meals, timing of bed and wake times, and smartphone use. Those who met inclusion criteria were invited for an in-person visit at Columbia University during which they underwent medical history screenings, anthropometric measurements of body weight, height, waist circumference, and blood pressure, and completed a morning-eveningness questionnaire to assess chronotype [24]. Participants were coached to download the free myCircadianClock (mCC) smartphone app to their iOS and Android devices and were instructed to log all eating occasions (>0 kcals) into the mCC app at the exact time of the eating event, without changing their eating habits, over 14 days. Eating occasions were logged via the in-app camera function to take a picture of the food or beverage or via text entries. Pictures of food or beverages were not used to calculate energy intake. The timestamp of every eating occasion allowed analyses of the temporal aspect of eating. Internal validation of the mCC app, with random pushes to verify entries, was completed, with a calculated error in the 10% range [3]. Additionally, the mCC app has been used in other clinical trials [3,19,20]. Participants were not provided instructions on total calories or dietary composition during the assessment period. Push notifications and in-app reminders were sent randomly several times per week to encourage app usage.

### 2.2. TEP Study

Temporal eating patterns (TEP) among adults with obesity was an ancillary pilot study to a behavioral weight-loss intervention with dietary guidance personalized to reduce postprandial glycemic response using a gut-microbiome-derived machine learning algorithm (Clincialtrials.gov NCT03336411). Participants were recruited from the New York University Langone Health electronic medical record (MyChart). Eligible participants were screened via phone and further screened in person to assess eligibility. Inclusion criteria for the TEP study included adults ages 18–80 years, BMI 27–42 kg/m^2^, pre-diabetic (HbA1c 5.7–6.4%) or early-stage type 2 diabetes (HbA1c <8% managed with lifestyle or lifestyle plus metformin), and with an estimated glomerular filtration rate >60 mL/min/1.73 m^2^. Participants were excluded from the ancillary study if they reported shift work or were unable to comply with logging self-reported meal times in the smartphone app. Additional information on exclusion criteria has been previously published [25]. Enrolled participants downloaded the free smartphone app (Personalized Nutrition Program, PNP) and were instructed to enter all eating occasions, including timing, into the app for up to 10 days. An individual from the research team contacted the participants daily by phone call or SMS text message to confirm the timing of each eating occasion entered into the app. The PNP app did not have an in-app camera feature; therefore, all eating occasions were entered manually. The PNP app has been used in a prior clinical trial [26]. One difference between the two apps (mCC vs. PNP) is the user interface. The mCC app provides feedback on mealtimes and eating windows, as it was designed to capture eating patterns. In contrast, the PNP app was developed to provide feedback on total energy intake, as well as relative intake from carbohydrates, proteins, and fats.

### 2.3. Metabolic Outcomes

Participants in the NY-TREAT pilot study were measured in a non-fasted state, either in the morning or in the afternoon, at the Columbia University Irving Medical Center Research Center. All measurements for participants in the TEP pilot study were conducted in the morning at the New York University Langone Health Clinical and Translational Science Institute, after an overnight fast of at least 8-h.

### 2.4. Anthropometrics

In the NY-TREAT study, height was measured to the nearest 1 cm, and body weight was measured in light clothing, without shoes, after voiding, to the nearest 0.1 kg, using a digital scale with a stadiometer (SECA 769 Seca GmBH & Co. KG, Hamburg, Germany). In the TEP study, height was measured to the nearest 1 cm using a portable stadiometer (SECA 213, Seca GmBH & Co. KG, Hamburg, Germany), and body weight was measured in light clothing, without shoes, to the nearest 0.1 kg, using a Stow-A-Weigh scale (Scale Tronix, Welch Allyn, Skaneateles, NY, USA). Among TEP study participants only, body composition (percent body fat and skeletal muscle mass) was measured using bioelectrical impedance analysis (InBody 270, InBody, Inc., Cerritos, CA, USA). Among NY-TREAT study participants, waist circumference was measured at the level of the umbilicus, and hip circumference was measured as the maximum circumference over the buttocks, with a tape measure, to the nearest 1 cm. Waist circumference and hip circumference were measured in triplicate, and the average of the 3 measurements was recorded; waist-to-hip ratio was calculated from these measures. In the TEP study, waist circumference and hip circumference were measured in duplicates using a Gulick tape (McKesson Medical-Surgical, Fairfield, NJ, USA), to the nearest 1 cm, and waist-to-hip ratio was generated from these measurements.

### 2.5. Blood Pressure

In the NY-TREAT study, systolic and diastolic blood pressures were measured manually by the study physician with a manometer (Welch Allyn PROPAQcs, Welch Allyn, Inc., Skaneateles Falls, NY, USA), following a minimum of 10 min of resting by the participant. In the TEP study, systolic blood pressure and diastolic blood pressure were measured following a 5-min, seated resting period using an automated blood pressure machine (Welch Allyn PROPAQcs, Welch Allyn, Inc., Skaneateles Falls, NY, USA).

### 2.6. Definitions of Temporal Eating Patterns

All eating-occasion times were aggregated from the two studies (TEP and NY-TREAT) and from these data the following variables were calculated: (1) *Eating occasions*. An eating occasion was defined as the consumption of food or beverage with caloric value (>0 kcal). The total number of eating occasions was defined as the sum of all eating occasions within a 24-h period. All eating occasions were included in a *metabolic day* if they fell between 04:00–23:59, based on a prior study [3]. Eating occasions between 00:00–03:59 were included in the previous day’s total. First and last eating occasions were defined as the local time (hh:mm) of the first and last caloric event, starting at 04:00. Eating occasions logged as coffee (e.g., iced, black), tea (e.g., green), water, seltzer water, or non-caloric event (i.e., vitamins, gum) were not included. (2) *Eating Window*. The 95% eating window was defined as the 95% interval of all eating occasions entered into the PNP app (TEP study) and mCC app (NY-TREAT study), as previously reported [3,25,26]. Self-reported eating window was generated as the interval between the first and last self-reported eating occasion in NY-TREAT participants only. (3) *Eating Midpoint*. Eating midpoint was defined as the median of all eating occasions. (4) *Within-person variability of eating occasions*. Irregularity of mealtimes was assessed by calculating the within-person variability of the number of daily eating occasions using the coefficient of variation (((standard deviation)/(mean)) × 100) for eating-occasion frequency and the timing of first and last eating occasions for each participant, using a minimum of 2 days.

### 2.7. Other Metabolic Outcomes (TEP Study Only)

Hemoglobin A1C (HbA1c) was analyzed using high-pressure liquid chromatography (Variant II Turbo analyzer, Bio-Rad Laboratories, Inc., Hercules, CA, USA). Fasting glucose and insulin were collected in lithium heparin tubes and analyzed with quantitative enzymatic assay hexokinase/G-6-PDH (Abbott Architect, Abbott Laboratory, Chicago, IL, USA) and chemiluminescent microparticle immunoassay (Abbott Architect, Abbott Laboratory, Chicago, IL, USA), respectively. Insulin levels were used in the Homeostatic Model Assessment (HOMA) to estimate steady-state beta cell function (HOMA-*β*) and insulin resistance (HOMA-IR; HOMA2, Diabetes Trials Unit, University of Oxford, Headington, Oxford, United Kingdom).

### 2.8. Statistical Analysis

Data were reported as median and interquartile range [IQR] unless otherwise noted. The Shapiro-Wilk test was run to assess normality of continuous variables. Participant characteristics were compared between the two studies (TEP vs. NY-TREAT) using independent samples, 2-sided *t*-tests for normally distributed continuous variables, Kruskal-Wallis tests for non-normally distributed variables, and Chi-squared tests for categorical variables. Pearson correlations were performed to analyze relationships between normally distributed variables, and Spearman correlations were performed for non-normally distributed variables. Linear regression models were performed to test the association between temporal eating-pattern variables and metabolic outcomes. The adjusted model included the covariates of age and gender. All data were described for the combined samples (TEP and NY-TREAT) and by gender. Eating-pattern variables were generated based on all eating occasions from aggregate data and from eating occasions on adherent days, where a minimum of 2 eating occasions were logged ≥5 h apart [3]. Bias, limits of agreement, and the plot of bias against the median 95% eating window and self-reported eating window were analyzed with a Bland-Altman plot. All statistical analyses were conducted using SPSS (SPSS version 23, IBM, Armonk, NY, USA), and figures and graphs were generated with GraphPad Prism (Prism 8.4.2, GraphPad Software, LLC, San Diego, CA, USA). The level of significance was set at an alpha of 0.05.

## 3. Results

Participant characteristics for the entire cohort and by study are shown in Table 1. A total of 106 participants were recruited between both study sites. The TEP study recruited and enrolled 42, with 35 included in the final analysis. Five participants did not provide data due to COVID-19-related issues, and two failed to record eating occasions in the smartphone app. The NY-TREAT study recruited 64 participants, and 50 completed the two-week assessment period. Eating occasions were inappropriately logged by one participant, whereby they did not log a minimum of two eating occasions ≥5 h apart; therefore, eating-pattern variables were generated from 84 participants. Small but significant differences in blood pressure and body circumferences between the two studies may be attributed to measurement methods but are most likely due to a worse metabolic phenotype in TEP participants, based on inclusion criteria.

The duration of the observation period was 14 days for NY-TREAT and 10 days for TEP. There was no significant difference between any of the eating-pattern variables by study type (Appendix A). A total of 3472 eating occasions were logged when the data from TEP and NY-TREAT were aggregated, with a total of 340 eating occasions logged on non-adherent days. Therefore, a total of 3132 eating occasions were included in the analysis of eating patterns for the combined sample. A total of 110 eating occasions (3.5%) were logged between 00:00 and 03:59 (Figure 1). Peak eating occasions occurred at 12:30, with additional peaks observed between at 18:00 and 19:00, which may indicate main meals for lunch and dinner, respectively. Assuming the metabolic day starts at 04:00, 29.7% of eating occasions were logged before noon and less than a quarter (18.2%) of all eating occasions were logged between 18:00 and 23:59. Eating patterns from all eating occasions are shown in Appendix A.

Table 2 details eating-pattern variables for the combined samples, by gender and by weekend vs. weekday for all eating occasions logged on adherent days. Eating window, stratified by the onset of nighttime fast, is depicted in Figure 1. The longest observed eating window was 18 h 58 min, and the shortest was 10 h 29 min. Over 50% of participants had an eating window >14 h/d, with only 13.1% of participants with an eating window <12 h/d. There were no significant differences between males and females for all variables. Comparing weekends vs. weekdays, participants had fewer eating occasions, delayed first eating occasions, and had shorter eating windows on weekends. Additionally, eating irregularity, characterized by within-person variability, was not significantly different between genders. However, within-person variability of the first reported eating occasion was lower on the weekends compared to the weekdays.

Eating-pattern variables were not significantly associated with body weight, BMI, waist circumference, hip circumference, or waist-to-hip ratio (Appendix A). There were no significant associations between 95% eating window and body weight, BMI, waist circumference, hip circumference, or waist-to-hip ratio (Appendix A) for the total sample or in the adjusted models (data not shown). A sub-analysis of only TEP participants (n = 33) found that the 95% eating window was not significantly associated with % body fat, after adjusting for age and gender (B = 19.2, 95%CI: −3.1, 41.5; p = 0.09). Additionally, glycemic outcomes collected from TEP participants found no association between 95% eating window and HbA1c, fasting glucose, fasting insulin, or HOMA-IR, except for HOMA-β (r = 0.297, p = 0.02; Appendix A).

## 4. Discussion

Quantifying baseline temporal eating patterns is central to selection of individuals who may benefit most from TRE and to demonstrate the efficacy of chrononutrition-based interventions. The aim of our study was to describe temporal eating patterns in adults with overweight or obesity and to assess the within- and between-person variability in eating patterns in this population. Our main finding was that >52% of our participants had a long eating window of >14 h/d, with the longest eating window of 18 h 58 min. We found no significant association between eating window and anthropometric measures. Our data also showed that adults with overweight or obesity reported approximately four eating occasions/day, with the peak number of eating occasions occurring around 12:30, and had their first eating occasion at 09:40 and last at 19:40. Additionally, weekend eating patterns exhibited fewer logged eating occasions, a delay in first eating occasion, a shorter eating window, and less within-person variability in the first eating occasion when compared to that of weekday eating patterns.

Our results show no association between eating window and anthropometric measures (i.e., waist circumference). Similar results were reported elsewhere in a sample of young adults with normal BMI [3]. Our findings may be due to a fairly homogenous sample with a small range of BMI. BMI is a poor representative of fat mass, especially in men [27]. Additionally, in our subgroup analysis, we found no associations between percent body fat and eating window, which have been reported by others [28,29]. A sample of adults with obesity found no association between body fat measured using dual-energy x-ray absorptiometry and eating window [29]. Collectively, our findings suggest no link between adiposity and eating window; however, it is unclear whether this relationship is independent of energy intake.

A decrease in the eating window with TRE is effective at reducing body weight [3,18,19]. For example, in a 12-week, 10 h TRE intervention, Wilkinson et al. found an approximately 28% reduction (−4.35 ± 1.32 h) in eating window from pre- to post-intervention, and [19] Chow et al. reported a significant decrease in the eating window of 5.5 ± 2.2 h following a self-defined eating window of 10 h [20]. As mentioned previously, the ability of TRE interventions to successful reduce body weight is likely the result of reduced total energy intake. Prior interventions using self-reported methods to measure energy intake estimate a reduction of approximately 20–30% in energy intake following a reduced eating window [3,18,19]. To our knowledge, there are no studies showing a decrease in weight loss due to a reduction in eating window (i.e., extended fast) independent of a decrease in energy intake.

The median eating window in our study was >14 h/d, which is similar to that of prior reports on this population using a similar definition [7]. Others have reported shorter eating windows [4,29,30,31], which may be the result of how eating window was defined. Typically, eating window is defined as an interval between the first and last eating occasions (hereafter interval method); however, this definition is prone to misreport actual mealtimes and unlogged (missed) meals. Furthermore, this method does not account for day-to-day variations in the time of first and/or last meal. For example, in a pilot study assessing an 8-h TRE intervention, Gabel et al. reported an average eating window of 11 ± 1 h in adults with obesity but acknowledged the inaccuracy, given the use of self-reports [30]. Comparable eating windows have been shown when generated from self-reports, such as with undergraduate and postgraduate students in Spain and Mexico [31] and a Dutch sample of adults with overweight and obesity, with a daily eating window of 12.3 ± 1.8 h [29]. Together, these studies demonstrate the possibility of misrepresentation of the habitual eating window when measuring the difference between first and last eating occasion.

Self-reported eating window maybe used as a screening tool for TRE interventions (e.g., “On average, at what time do you consume your first meal?”). However, self-reported eating occasions are prone to underreporting, and the association between self-reported and measured eating window using an app is unclear [32,33]. The use of the 95% eating window derived from several days of food logs may avoid misreporting and is less prone to measurement error, as it considers all eating occasions. The limitation of self-reporting is well illustrated by our finding in NY-TREAT participants, with a large bias (−1 h 50 min ± 2 h 56 min) and wide limits of agreement compared to the measured 95% eating window using the mCC app (Appendix A). Application of qualitative screening measures (e.g., self-report, questionnaire) to characterize eating patterns may lead to the unintended exclusion of participants who may benefit from an intervention that alters eating patterns. Without a better understanding of habitual eating patterns, it is challenging to discern how chrononutrition-based interventions change eating patterns, aside from altering the feeding/fasting periods. The discrepancy in eating windows between studies may also be attributed to differences in the populations studied, such as young versus middle-aged adults or between large urban centers, like ours in New York City, versus more a rural environment.

Eating patterns are impacted by the day of the week, as evidenced by a shift in the sleep-wake cycle, resulting in circadian desynchrony (i.e., social jetlag) [34]. In a cross-sectional analysis of middle-aged adults, participants with social jetlag >1 h (that is, the absolute difference between mid-sleep time on weekends and weekdays) ate their breakfast, early afternoon snack, and dinner at later times and had a longer eating duration than those without social jetlag [35]. In our study, we observed a roughly 42 min delay in the median first eating occasion on the weekend compared to the weekdays. However, there was no difference in the timing of the last eating occasion between weekends and weekdays. As a result, the overall eating window on weekends was shorter compared to that on weekdays. Similar findings comparing weekends and weekdays have been previously reported [3]. Differences in eating patterns between weekdays and weekends may be due to lower adherence to eating-occasion reporting. While we did not measure sleep, we speculate that the delayed first eating occasion on weekends maybe the result of an altered sleep patterns, whereby bed and wake times are also delayed.

Additionally, there was greater within-person variability of the first eating occasion on weekdays compared to weekends, despite a delay in the timing of the first eating occasion on weekends. Based on these patterns, it appears adults with obesity may skip or delay their first eating occasion to a greater degree during the week compared to the weekend but are rigid in the timing of their last eating occasion throughout the entire week. Greater within-person variability in first eating occasion may be due to other lifestyle factors, such as employment or family responsibilities (i.e., childcare). Future studies should assess the within-person variability of eating patterns over a longer period (>4 week) to further elucidate the relationship with biological outcomes (e.g., body-weight variability).

Strengths of the present study include recording of eating occasions for up to 14 consecutive days and the use of smartphone apps to track and quantify eating patterns; however, there were some limitations. While two different smartphone apps were utilized to collect eating-pattern data, there were no between-study differences in eating-pattern variables. The patient-facing data and information provided with smartphone apps may influence an individual’s ability to record an eating occasion, dependent on the individual’s technological literacy and experience. Some smartphone apps are easier to navigate and access embedded information with than others, thus reducing barriers to logging. Temporal eating patterns are defined not only by timing but also by distribution of energy intake, which we were unable to quantify in the TEP and NY-TREAT studies. Prior TRE studies assessing the impact of TRE on weight loss report a 20–30% reduction in self-reported energy intake accompanying a reduction in the eating window [3,18,30]. Relying on self-reported meal timing assumes that participants log all eating occasions in real time. However, misreporting and forgetfulness may occur. Considering missed eating occasions may be associated with the outcomes of interest, this could have introduced not only random but also systemic bias into our analysis. This was mitigated by built-in reminders and random text-message confirmations of mealtimes in the TEP study and random text messages and reminder messages built into the mCC app (NY-TREAT study). However, these characteristics were not integrated into the smartphone PNP app used by TEP participants. There is an urgent need to develop objective methods of measuring meal content and timing (i.e., wrist-motion, chewing) that do not rely on individual recall, especially given that there is currently no gold standard in eating-pattern quantification. Additionally, a critical component that anchors food intake behavior is sleep. Unfortunately, wake and bedtimes, as well as sleep duration, were not available in either study. Inclusion criteria for both studies included overweight and obese participants; therefore, we are unable to make eating-pattern comparisons with normal-BMI participants.

## 5. Conclusions

In summary, we found that the majority of adults with overweight or obesity have an eating window >14 h/d, with only 13.1% of participants with an eating window <12 h/d. Anthropometric measures (e.g., WHR) were not associated with eating window; however, future studies are required to confirm these findings using more precise measures of body composition in a larger, more diverse sample, including measures of energy intake. We confirm prior findings showing differences in eating patterns between weekdays and weekends; in particular, we found greater within-person variability for the first eating occasion on weekdays compared to weekends. Moreover, the timing of the last eating occasion was consistent throughout the week. We encourage future TRE or intermittent fasting interventions to use the 95% eating window to define an individual’s eating duration. The findings of this study, specifically long eating windows and differences in eating patterns between weekdays and weekends, highlight the importance of characterizing baseline eating patterns prior to initiating chrononutrition-based interventions.

## Figures and Tables

**Figure 1 nutrients-13-04485-f001:**
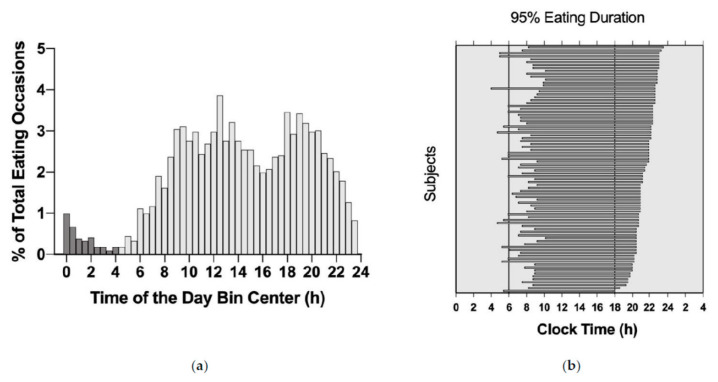
Percent of total eating occasions over 24 h. (**a**) % Total eating occasions in 30 min bins are represented by individual bars. A total of 97.3% of all eating occasions were logged between 04:00 and 23:59 (light grey bars), and 3.5% of all eating occasions were logged between 0:00 and 03:59 (dark grey bars). Peak logging of eating occasions occurred around 12:30. Aggregate data from the TEP and NY-TREAT studies were included. 95% Eating Window for Aggregated Data from Participants in the TEP and NY-TREAT studies. (**b**) The 95% eating duration shown in the order of the late (**top**) to early (**bottom**) nighttime fasting onset time. Each bar represents the 2.5–97.5 percentile for an individual. Solid black lines at 06:00 and 18:00 represent a reference 12 h 0 min eating window.

**Table 1 nutrients-13-04485-t001:** Participant Characteristics.

**Variable**	**All (*n* = 85)**	**TEP (*n* = 35)**	**NY-TREAT (*n* = 50)**	***p*-Value**
Age, years	56 ± 19	58 ± 16	50 ± 24	**0.009**
Sex, % (*n*)				
Male	28.2 (24)	42.9 (15)	18.0 (9)	**0.024**
Female	71.8 (61)	57.1 (20)	82.0 (41)	
Ethnicity, % (*n*)				
Non-Hispanic	70.6 (60)	82.9 (29)	62.0 (31)	0.066
Hispanic	29.4 (25)	17.1 (6)	38.0 (19)	
Race				
African American	30.6 (26)	57.1 (20)	56.0 (28)	0.127
Caucasian	56.5 (48)	22.9 (8)	36.0 (18)	
Other	9.4 (8)	11.4 (4)	8.0 (4)	
Unknown	3.5 (3)	8.6 (3)	0 (0)	
Height (cm)	165.9 ± 7.9	166.8 ± 9.4	165.3 ± 6.7	0.408
Weight (kg)	91.4 ± 22.3	91.4 ± 19.4	91.1 ± 26.2	0.489
BMI (kg/m^2^)	31.8 ± 8.0	32.2 ± 4.4	31.0 ± 10.5	0.372
SBP (mmHg)	122.0 ± 19.0	126.5 ± 14.3	113.5 ± 20.0	**0.005**
DBP (mmHg)	75.3 ± 9.1	75.4 ± 8.4	75.2 ± 9.7	0.922
WC (cm)	103.4 ± 12.2	107.3 ± 10.5	100.7 ± 12.6	0.011
HC (cm)	108.5 ± 15.3	111.5 ± 11.9	106.7 ± 19.3	**0.018**
WHR	0.9 ± 0.1	0.9 ± 0.1	0.9 ± 0.1	0.303
Percent Body Fat (%)	-	38.9 ± 7.8	-	-
HbA1c (%)	-	5.9 ± 0.5	-	-
Fasting Glucose (mg/dL)	-	95.0 ± 19.0	-	-
Fasting Insulin (mIU/L)	-	10.0 ± 7.0	-	-
HOMA-*β*	-	90.3 ± 51.6	-	-
HOMA-IR	-	1.35 ± 0.9	-	-
Chronotype, % (*n*)				
Morning type	18.8 (16)	0 (0)	32.0 (16)	**0.004**
Intermediate	48.2 (41)	54.3 (19)	44.0 (22)	
Evening type	10.6 (9)	11.4 (4)	10.0 (5)	

BMI, body mass index; SBP, systolic blood pressure; DBP, diastolic blood pressure; HbA1c, glycated hemoglobin; HOMA-*β*, homeostasis model assessment for beta-cell function; HOMA-IR, homeostasis model assessment for insulin resistance. Values are reported as mean ± SD, except age, BMI, SBP, SBP, HC, fasting glucose, and fasting insulin HOMA-*β* and HOMA-IR, which were not normally distributed and are reported as median ± interquartile range; significance in bold; Chronotype, *n* = 66.

**Table 2 nutrients-13-04485-t002:** Eating patterns and within-person variability from eating occasions on adherent days.

Variable	All (*n* = 85)	Male (*n* = 24)	Female (*n* = 61)	*p*-Value Male v Female	Weekday	Weekend	*p*-Value Weekday v Weekend
Eating Occasions (#/day)	4.0 [3.1–4.6]	4.0 [3.5–4.5]	4.0 [3.1–4.6]	0.944	3.9 [3–4.6]	3.5 [2.7–4.5]	0.031 *
First Eating Occasion, (hh:mm)	9:41 [08:43–10:26]	9:34 [08:36–10:35]	9:41 [08:53–10:19]	0.988	9:29 [08:34–10:17]	10:10 [08:51–11:02]	0.013 *
Last Eating Occasion, (hh:mm)	19:41 [19:02–21:05]	20:23 [19:02–21:32]	19:37 [19:02–20:39]	0.309	19:29 [18:58–20:57]	19:54 [19:04–21:28]	0.208
95% Eating Window, (h, min)	14 h 38 min [12 h 59 min–16 h 49 min]	14 h 33 min [13 h 1 min–17 h 9 min]	14 h 38 min [12 h 59 min–16 h 14 min]	0.604	14:11 [12:50–15:48]	12:31 [10:34–14:39]	<0.0001 *
Eating midpoint, (hh:mm)	14:13 [13:21–14:55]	14:30 [13:58–14:53]	14:02 [13:20–14:55]	0.333	14:13 [13:14–15:09]	14:06 [13:02–15:42]	0.358
Within-person variability eating occasion frequency, (%CV)	28.2 [23.0–35.0]	27.0 [22.5–29.0]	32.0 [23.0–37.0]	0.101	27.5 [22.0–35.5]	28.0 [18.5–41.0]	0.401
Within-person variability First Eating Occasion, (h, min)	4 h 29 min [3 h 21 min– 6 h 17 min]	4 h 24 min [3 h 28 min– 5 h 9 min]	4 h 55 min [3 h 19 min– 6 h 18 min]	0.524	4 h 47 min [3 h 4 min– 6 h 31 min]	2 h 48 min [1 h 26 min– 4 h 57 min	0.0002 *
Within-person variability Last Eating Occasion, (h, min)	2 h 3 min [1 h 38 min– 2 h 46 min]	2 h 0 min [1 h 34 min– 2 h 48 min	2 h 3 min [1 h 42 min– 2 h 40 min]	0.773	1 h 55 min [1 h 36 min– 2 h 45 min]	1 h 39 min [0 h 47– 2 h 43 min]	0.263

%CV, percent coefficient of variation; significance * *p* < 0.05. Linear regression was performed to test the association between temporal eating-pattern variables and metabolic outcomes. The adjusted model included covariates of age and gender. All data were described for the combined samples (TEP and NY-TREAT) and by gender. Eating occasions were not normally distributed and were therefore reported as median [interquartile range]; eating pattern analyses were generated based on eating events that included adherent days (>2 eating occasions logged >5 h apart). First and last eating occasions, as well as eating midpoint, are reported as local times (hh:mm).

## Data Availability

Not applicable.

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
