# Peer review of "Temporal Eating Patterns and Eating Windows among Adults with Overweight or Obesity"

_nutrients, 2021, doi:10.3390/nu13124485_

Round 1

Reviewer 1 Report

The manuscript suffers from numerous grammatical errors that make reading difficult. Please correct these.

The abstract requires some sort of conclusion. What do these data represent?

Line 50: what percentage is < 12 h?

Line 97: please describe the app. Has it been validated for use in research studies?  Following this comment – please describe the other app. What are the important differences between the two?

To facilitate understanding, I recommend writing early morning times with a leading zero, e.g., 04:00.

Line 165: I don’t understand the definition of the 95% eating window that is provided. Please explain further. 95% of what?

Paragraph starting at line 284: this paragraph starts off by cataloging studies that show no association but concludes by explaining why there should be. This is confusing, and there is no conclusion to the paragraph.

Line 303 doesn’t make sense as written.

Paragraph starting at line 300: the topic sentence of this paragraph starts states that there are studies that observe that reducing the eating window leads to weight loss. Mid-way through, the subject changes to studies that don’t observe these findings. These changes in topic midway through make the discussion extremely difficult to read.

Line 330: 12 hours is comparable to 14 hours? Does the “interval method” refer to your method or the method described in the paragraph?

Line 335: self-reported eating occasions are underreported – this should be cited.

Line 336: please restructure this sentence and correct the grammar as it is difficult to understand. I don’t understand how it is less prone to error by taking into account all eating occasions than considering the first and last?

Paragraph starting at 333 – again, a variety of topics not represented by the topic sentence. Self-reported windows vs qualitative screening tools – do you mean that self-reported windows are qualitative screening tools?  The last part of this paragraph talks about: qualitative screening measures then the need to understand habitual eating patterns, then the discrepancy in eating windows  - I do not understand the logic of jumping from one topic to the next.

Line 363: if the last eating occasion did not differ by weekday vs weekend, how do you conclude that bedtime was later?

Line 370: “Future studies should assess the within-person variability of eating patterns over a longer period of time to further elucidate the relationship with biological outcomes (e.g., body weight variability).” But didn’t you state there was no relationship between this and BMI? How much longer would this need to be studied?

Line 401: please replace “adults” with “participants” given the small scale of the study.

Last line: “These data highlight the importance of characterizing baseline eating patterns prior to initiating chrononutrition-based interventions.” What about these data highlight that importance?

Author Response

We want to send a heartfelt thank you to the reviewer for taking the time and effort to read and review our manuscript. We greatly appreciate their comments and suggestions.

Reviewer 2 Report

Chrononutrition and temporal eating patterns (TEP) are now a research hot topic. In this study, the authors aimed to identify temporal eating patterns in a sample population of adults with overweight and obesity, where time-restricted eating (TRE) could modulate cardiometabolic health outcomes. The aim, design, and methods are well defined, and the strengths and limitations of the study are well discussed. These finding results could be improved with a large sample, to provide rational for future chrononutrion-based interventions on obesity and metabolic chronic conditions. 

Author Response

We thank the reviewer for taking the ime to read and offer comments on our manuscript. We agree with them that a much larger sample of adults should be studied. In particular, we need to examine adults with BMIs <25 kg/m2.